# Assessing the Impact of Antimicrobial Resistance Awareness Interventions Among Schoolchildren in Bangladesh

**DOI:** 10.3390/antibiotics14100979

**Published:** 2025-09-29

**Authors:** S. M. Sabrina Yesmin, A. T. M. Golam Kibria Khan, Umme Habiba, S. M. Shanzida Yeasmin, Mohammad Delwer Hossain Hawlader

**Affiliations:** 1Directorate General of Drug Administration, Dhaka 1212, Bangladesh; 2Department of Pharmacy, Faculty of Sciences and Engineering, East West University, Dhaka 1212, Bangladesh; keka077@gmail.com; 3Marketing Department, Northern University Bangladesh, Dhaka 1213, Bangladesh; shanzidayeasmin15@gmail.com; 4Department of Public Health, North South University, Dhaka 1229, Bangladesh; mohammad.hawlader@northsouth.edu; 5NSU Global Health Institute (NGHI), North South University, Dhaka 1229, Bangladesh

**Keywords:** antimicrobial resistance, schoolchildren, awareness

## Abstract

**Background**: Antimicrobial resistance (AMR) is a critical global health issue. Like other low- and middle-income countries, the misuse of antimicrobial medicine, including widespread self-medication, exacerbates AMR in Bangladesh. Making future generations aware of AMR through educational interventions is an effective tool in combating AMR. This research focuses on understanding the effects of AMR awareness interventions on the knowledge, attitudes, and behaviors of the schoolchildren in the selected district of Bangladesh. **Methods**: In this study, 241 students of the 12- to 16-year-old age group participated in a two-day program. The programs include four hours of activities, including reading comics and coloring books, presentations, quizzes, and watching an animation about AMR on the first day, followed by an art competition on the second day. To assess changes in knowledge earlier and after the intervention, pre- and post-tests were conducted. **Results**: This pilot study demonstrates that using age-appropriate interactive educational tools can significantly improve students’ knowledge about AMR, showing a mean difference of 1.28 (*p* < 0.001). The regulatory step of the Directorate General of Drug Administration, incorporating red identification marks on antibiotic packaging, makes it easier and shows that 93.36% of students could identify antibiotics, which helps them to be aware of these types of medicines. Interventions were equally effective for boys and girls and science and commerce students, and these helped participants recognize the inappropriate practices of antibiotic use in their daily lives. **Conclusions**: This study identified the importance of incorporating AMR issues into the educational curriculum to address AMR for future generations.

## 1. Introduction

Antimicrobial resistance (AMR) is one of the top 10 global public health risks, which requires urgent multisectoral action to achieve the Sustainable Development Goals (SDGs) [1]. Without effective antimicrobials, the success of modern medicine in treating infections, including major surgery and cancer chemotherapy, would be at increased threat. Lack of mindfulness and legislation in developing countries is one of the reasons for the arising antimicrobial resistance issue [2]. From a public health viewpoint, this situation needs special attention because antibiotic-resistant bacterial infection caused an estimated 1.27 million deaths in 2019 [3]. AMR is increasing over the years in Bangladesh owing to its poor healthcare standards, along with the misuse and overuse of antimicrobials, thus posing a regional and global threat. Moreover, aggressive and unethical marketing practices of pharmaceutical companies are adding more to the problem, as a proper regulatory regime is lacking to oversee this vast market [4]. A study by Saha S. et al. [5] shows that 22.7% of all medicines are sold without a prescription in Bangladesh, and the highest-selling medicines are antibiotics. A study by Ahmed S.M. et al. [6] shows that the physician-to-population ratio of Bangladesh is 5 per 10,000, and another study by Mahumud R.A. et al. [7] shows that the out-of-pocket expenditure on medicine in Bangladesh accounted for 61% of total out-of-pocket healthcare expenditure. Hence, the poor people often have to choose between going untreated and spending a large amount of money on medicines [5]. As a result, self-medication is much more popular than healthcare seeking in the formal health system in Bangladesh.

Social and behavioral drivers of inappropriate antibiotic use have been identified as one of the key contributing factors to the emergence of AMR. Enhancing the understanding of the broader public of the causes and consequences of AMR and their role in minimizing antibiotic misuse is considered to be an important component of an effective and optimal public health response [8]. Global consultation meetings on awareness raising on antimicrobial resistance, organized by WHO, identified one of the targeted audiences for AMR awareness work as children/students/youth and also focused on the importance of targeting schoolchildren and students from primary school to university [9].

WHO emphasizes the investment in AMR education for children and adolescents, as it will impact the prudent use of antibiotics in the future for themselves, their families, and their communities [10]. However, most of the countries do not emphasize this issue, as the Quadripartite Global Database for Tracking AMR Country Self-assessment Surveys (TrACSS) [11] found that among 177 countries, 78% of the primary and secondary academic children do not receive education on AMR. To draw the attention of policymakers, our study has been conducted. Interventions aimed at schoolchildren and their parents could be key to addressing AMR, given their effectiveness in improving knowledge and attitudes and changing behaviors regarding responsible or rational use of antibiotics. A study conducted in Ghana had observed that picture drawing had significant effects (both positive and negative) on schoolchildren’s AMR knowledge, attitudes, and beliefs studied [12].

Awareness of AMR for future generations is an important mandate of the Directorate General of Drug Administration (DGDA), Bangladesh. DGDA has taken different AMR awareness initiatives, including regulatory intervention of incorporating red identification marks on the packaging materials of antibiotics for public awareness and easy identification of antibiotics [13]. The AMR school campaign in Cox’s Bazar by DGDA suggests that the AMR comic book and art competition were well accepted by schoolchildren [14]. However, studies assessing the impact are lacking in Bangladesh.

Hence, this pilot study is an attempt to explore whether different modes of effective communication tools can bring out positive social behavioral changes among schoolchildren. We conducted this study in order to evaluate the impact of AMR awareness interventions, measuring changes in knowledge, attitudes, and practices, determining the factors influencing effectiveness, and providing recommendations for future interventions. With these objectives, this study aims to contribute to the field of antimicrobial resistance awareness and inform strategies for addressing this serious issue among schoolchildren.

## 2. Results

Table 1 shows the socio-demographic characteristics of the respondents (N = 241). The majority of participants were aged 14–16 years (88.4%), with males comprising nearly two-thirds (62.7%). Most belonged to class 8 (46.9%), and Islam was the predominant religion (97.5%). Fathers were commonly engaged in business (28.2%) or the armed forces (24.1%), while most mothers were housewives (75.5%). Almost half of the students (46.6%) had two siblings.

Table 2 presents the paired *t*-test comparing pre- and post-intervention effects. Knowledge scores significantly increased after the intervention (mean difference = 1.28, *p* < 0.001), suggesting that the educational program was effective in improving participants’ understanding. This improvement was statistically highly significant (<0.001).

Figure 1 shows the percentage of correct answers by question for pre- and post-intervention. Correct responses increased across almost all questions, with the most notable improvements in items related to infection prevention and antibiotic use, indicating that the intervention addressed key knowledge gaps.

Table 3 shows the one-way Analysis of Variance (ANOVA), which indicates that the improvement in knowledge was consistent across gender, class, subject, religion, and parental occupation, as none of these factors showed significant differences (*p* > 0.05). This suggests that the intervention was effective regardless of socio-demographic background.

Figure 2a shows that before the intervention, most participants (60.2%) had never heard of AMR. After the intervention, awareness rose dramatically, with over 93% reporting familiarity with AMR.

Figure 3 shows the sources from which participants learned about AMR before the interventions, with school/teacher and media/advertisement being the main sources.

Table 4 shows the linear mixed effect model, which expresses no significant effects of gender or educational background (science vs. commerce) on improving knowledge about AMR after the interventions, as interventions show equal effects across all groups.

Figure 4 shows children’s illustrations revealed their grasp of how AMR spreads, with common themes including inappropriate antibiotic use, environmental contamination, and human-to-human transmission. These qualitative findings support the quantitative results by showing that the intervention enhanced both awareness and conceptual understanding. 

## 3. Discussion

Education and mindfulness raising on AMR are important tools in global health policy for changing public behavior and combating AMR [15]. There is a lack of AMR education and behavioral change intervention globally, and a systematic review by Fuller W. et al. [16] suggested an urgent need for context-specific AMR educational and behavioral change interventions.

The decision to administer antibiotics without a prescription is often determined by adults, particularly parents, rather than the children themselves. Even when children are reluctant, parental authority can lead to the use of antibiotics, highlighting the importance of involving both mothers and fathers in AMR educational programs to ensure consistent messages at home. In addition, children and adolescents have a direct influence on their families and communities. AMR through education was our focus to mitigate this problem for future generations, as students can develop a better understanding of antimicrobial use and disposal through interactive education [17]. Educating parents alongside children can reinforce appropriate practices and reduce unnecessary or inappropriate antibiotic use. Furthermore, restricting over-the-counter access to antibiotics is supported by the Drug and Cosmetics Act 2023, which imposes a penalty of BDT 20,000 (approximately USD 165) for selling antibiotics without a prescription in Bangladesh, helping to minimize misuse [18].

In our study, pre- and post-surveys show a significant knowledge upgradation among schoolchildren aged 12 to 16 through interventions, showing a mean difference of 1.28 (*p* < 0.001). The findings are consistent with global evidence that early education on AMR can influence future behaviors and promote responsible antibiotic use within communities [19].

Incorporating red identification marks on antibiotic packaging [13], which was a regulatory step of DGDA, represents a unique feature of this program, making it easier to identify antibiotics even for children (in the post-survey, 93.36% could identify antibiotics).

Self-medication with antibiotics, weak antibiotic stewardship programs, insufficient diagnostic facilities, and poor infection prevention and control (IPC) are common problems in low- and middle-income countries like Bangladesh [20]. Ordinary people cannot give up self-medication because antibiotics are easily available in pharmacies, shopkeepers sell them even without a prescription, and antibiotics cure diseases quickly [21]. Moreover, since out-of-pocket expenditure is very high in Bangladesh, people often do not want to visit the doctor for common infections, and saving money is a priority for them. Moreover, doctors or medical facilities are not easily available in remote areas [22]. In addition, inappropriate antibiotic use in hospital settings also contributes to antimicrobial resistance, indicating that targeted training for physicians, nurses, and other healthcare providers is essential to address AMR comprehensively at both community and clinical levels [23]. On the basis of this situation, the questionnaires were developed, which reflected well-known practices of participants’ families. This study shows that interventions helped participants recognize the inappropriate practice of taking antibiotics in their daily lives. Examples include knowledge improvement about sharing antibiotics with family members (21.2%), not completing the full course once they feel cured (8.3%), taking antibiotics for mild infections (e.g., fever, diarrhea, etc.) instead of natural remedies (6.6%), and even taking antibiotics from the pharmacy without a doctor’s prescription (11.2%).

This study also demonstrated the knowledge improvement about AMR. For example, participants understood that if antibiotics become ineffective over time for some individuals, they will also become ineffective for people around them (33.6%). That indicates the interventions were effective in understanding the critical issue of AMR. Similar outcomes were reported in a study conducted by Cebotarenco N. et al. [24] in Moldova, which showed a positive impact in reducing antibiotic use for cold and flu by applying the behavioral change interventions.

As antibiotics are very popular, and before interventions, participants (15.4%) believed that any antibiotic could cure all types of infections. The comic book “*Tinu Minu and Super Bug*” helped them to better understand this critical issue, as the story of this book highlights the harmful effects of self-medication with antibiotics for any kind of infection (knowledge improved by 11.3%).

AMR is a microbiological issue and is generally less familiar to students without a science background, but it is equally important for everyone [15]. Based on this study, science and commerce students were observed to enhance similarly after the interventions (estimate = 0.35, 95% CI: −0.02–0.73, *p* = 0.069), which means there was no statistically significant difference between the two groups. This suggests that tools such as comic books, animations, coloring books, and art competitions not only engaged students but also simplified complex microbiological and pharmacological concepts, making them easier to understand and remember.

Despite the potency of our study, it has some limitations. The socio-demographic composition was imbalanced, with a higher proportion of male and predominantly Muslim participants, which may limit the generalizability of the findings. The age range of participants was restricted to 12–16 years, representing a critical stage for shaping health behaviors; however, future studies should include broader age groups. Although the intervention emphasized judicious antibiotic use, there is a potential risk that excessive caution could lead to underuse of antibiotics when clinically necessary, highlighting the need for balanced messaging in future programs. Finally, it cannot be determined which component—animation, comic, coloring book, or art competition—contributed most to the observed effects. Furthermore, future studies should quantify the impact of each intervention and include participants from both urban and rural settings. In addition, involving parents in future interventions could further reinforce learning at home and promote responsible antibiotic use within families, enhancing the overall effectiveness of AMR education programs. A policy brief by Waswa J.P. et al. [25] indicated the significance of the integration of AMR content in primary and secondary school textbooks. In Bangladesh, the findings of our study have been presented in a round table meeting with policymakers, including the Ministry of Health and Family Welfare (MOHFW) [26], National Curriculum and Textbook Board [27], Directorate of Secondary and Higher Education [28], Directorate General of Health Services [29], and DGDA in October 2023. As a result, MOHFW requested the Ministry of Education to incorporate AMR topics in the secondary education curriculum of Bangladesh in December 2023 [30].

## 4. Materials and Methods

### 4.1. Data Collection

This pilot study was conducted in a school in Chittagong district, Bangladesh, targeting children aged 12 to 16 years, considering the local context and the typical ages of high school students in Bangladesh. According to UNICEF’s age categories, the targeted group is considered adolescents, who are particularly well-suited for counseling and behavioral change interventions targeting the future generation [31]. Primary-school-level students were not included in this study. Pre- and post-surveys were conducted to assess changes in knowledge before and after the intervention. Before data collection, written informed consent was obtained from all participants’ parents or guardians or class teachers, as the participants were adolescents. The participants were assured that their data would be kept confidential and only used for research purposes.

The program spanned two days: the first day consisted of four hours of activities, including reading comics and storybooks, presentations, and animation about AMR, while the second day was dedicated to an art competition. A total of 241 students participated. All students completed a pre-survey before the intervention and a post-survey afterward to measure learning outcomes. The survey questionnaires included 11 questions and are attached as Appendix A.

### 4.2. Sample Size Calculation

Using Cochran’s formula, the sample size for this study was calculated as follows:$$n=\frac{Z^{2}\times p\times (1-p)}{e^{2}}$$

Using a 95% confidence level (Z = 1.96), proportion (p) was set at 88.9% [12], and with a margin of error (e) of 4%, the initial sample size requirement was 237. Finally, data were collected from all 241 subjects.

### 4.3. Interventions

Following the WHO’s recommendation for AMR prevention and education in school, we developed interventional learning materials like comic books, coloring books, animations, and presentations. These materials focused on topics including understanding microorganisms, the basics of antimicrobials, what AMR is, and its impact. The main causes of AMR are prevention of infection and AMR, responsible use of antimicrobials, global and local responses to AMR, the role of everyone in the response to AMR, and future challenges and exploration through the interventions [10].

A pre-survey was conducted at the inception of the program. After the pre-survey, the day 1 program was designed in 5 parts, including a presentation, comic book reading, watching an animation, coloring book reading, and a quiz with a prize. Day 2 was dedicated to an art competition, followed by a prize-giving ceremony. A post-survey was conducted after the ceremony. Interventions are described in Table 5.

The animation of the comic book “*Tinu Minu and Super Bug*” is attached as Appendix A, the actual comic book “*Tinu Minu and Super Bug*” is attached as Appendix A, and the coloring book “*Invention of Penicillin*” is attached as Appendix A.

### 4.4. Data Analysis

Quantitative data collected from the participants were analyzed using Stata version 17. Paired *t*-tests for pre- and post-interventions, linear mixed effects models, and ANOVAs were conducted, and *p* < 0.05 was considered the threshold for statistical significance.

## 5. Conclusions

This study has shown that children are a very effective target population for creating awareness about antimicrobial resistance. If children can be taught about the dangers of antimicrobial resistance from childhood through textbooks, they will take antibiotics with awareness when they grow up. This will make future generations become more aware of risks associated with self-medicating or taking antibiotics without a doctor’s advice. Currently, where it is difficult to make people aware of different professions together and implement them, which requires a lot of manpower, money, and the involvement of print and electronic media, it is possible to easily create awareness among the entire nation by including the issue of antimicrobial resistance in children’s textbooks. Although this will take some time, it is expected to be very effective.

## Figures and Tables

**Figure 1 antibiotics-14-00979-f001:**
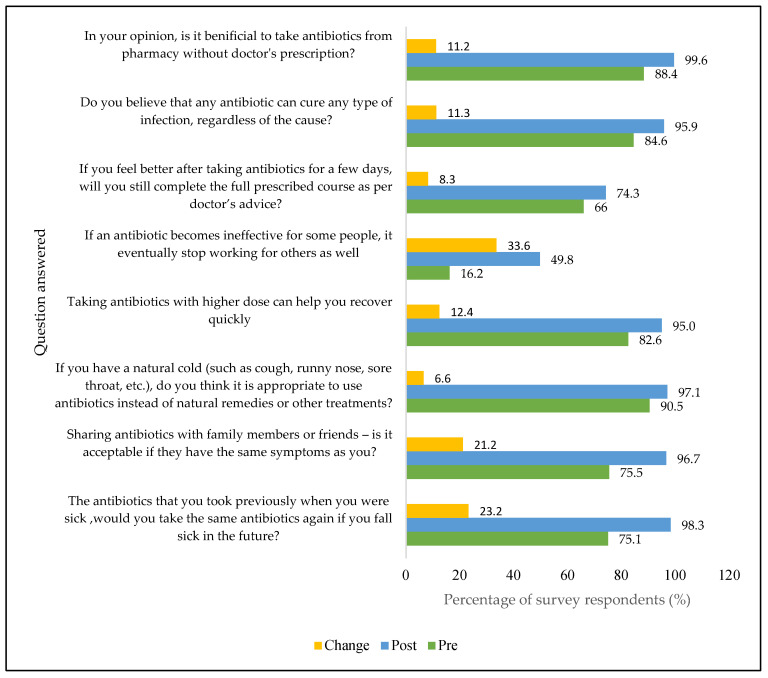
Percentage of correct answers categorized by question.

**Figure 2 antibiotics-14-00979-f002:**
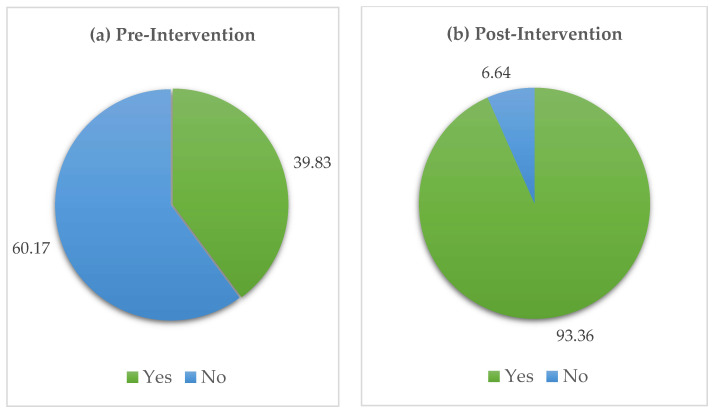
Percentage of participants who have heard about antimicrobial resistance.

**Figure 3 antibiotics-14-00979-f003:**
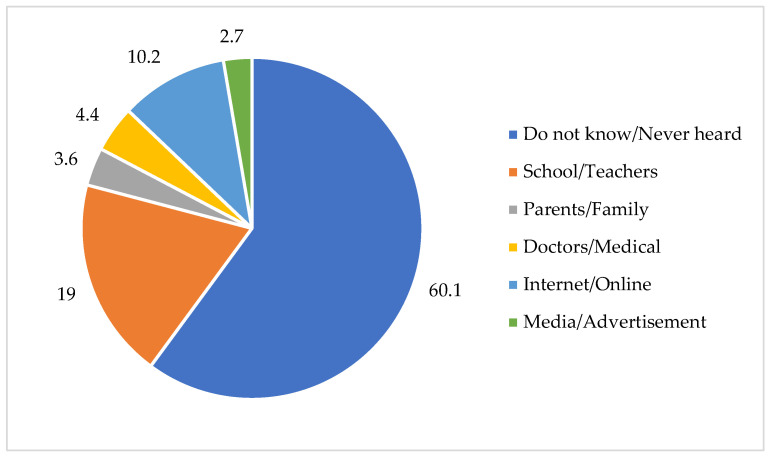
Percentage of sources from which participants learned about AMR (pre-survey).

**Figure 4 antibiotics-14-00979-f004:**
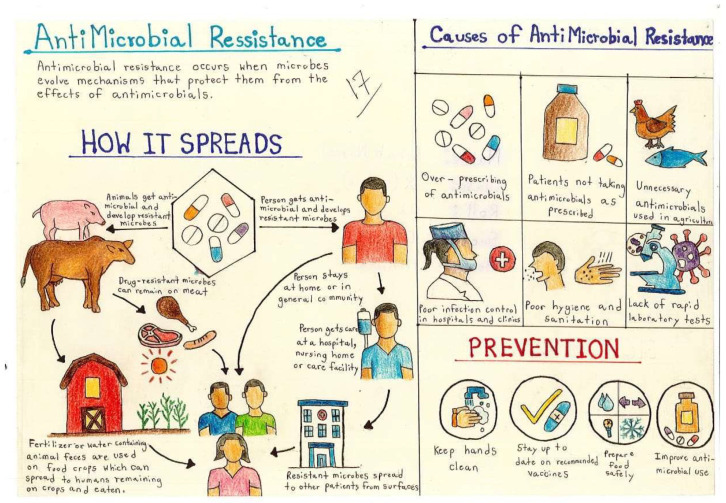
Children’s drawings about the spread of AMR.

**Table 1 antibiotics-14-00979-t001:** Socio-demographic characteristics of the surveyed participants.

Variables	n (%)
Age	
12	2 (0.8)
13	26 (10.8)
14	81 (33.6)
15	64 (26.6)
16	68 (28.2)
Sex	
Female	90 (37.3)
Male	151 (62.7)
Class	
8	113 (46.9)
9	76 (31.5)
10	52 (21.6)
Subject	
Business Studies	43 (17.8)
Science	85 (35.3)
Missing *	113 (46.9)
Religion	
Islam	235 (97.5)
Hindu	4 (1.7)
Buddhist/Buddhism	2 (0.8)
Father’s Occupation	
Armed Forces	58 (24.1)
Doctor	9 (3.7)
Teacher	19 (7.9)
Professionals (Engineer/Lawyer/ Banker)	24 (10.0)
Private Service	43 (17.8)
Govt. Officer	11 (4.6)
Business	68 (28.2)
Others	9 (3.7)
Mother’s Occupation	
Housewife	182 (75.5)
Teacher	27 (11.2)
Doctor	11 (4.6)
Others	21 (8.7)
Siblings	
0	3 (1.3)
1	58 (24.8)
2	109 (46.6)
3	53 (22.7)
4	9 (3.9)
5	2 (0.9)
Missing	7 (2.9)
Total	241

* The “Business Studies” and “Science” categories are for classes 9 and 10. Class 8 does not have these categories.

**Table 2 antibiotics-14-00979-t002:** Paired *t*-test for pre- and post-intervention outcomes.

Timepoint	Observations	Mean (Standard Deviation)	Difference	*p*-Value
Pre-intervention	241	5.79(1.65)	1.28	<0.001
Post-intervention	241	7.07(1.02)		

**Table 3 antibiotics-14-00979-t003:** Analysis of Variance for pre- and post-intervention outcomes.

	Degrees of Freedom	Sum of Squares	Mean Square	F Value	*p*-Value
Pre- and Post-intervention	1	36.2	36.16	19.244	<0.001
Gender	1	6.6	6.64	3.531	0.061
Class	2	0.8	0.42	0.224	0.533
Subject	1	6.9	6.934	3.059	0.081
Religion	1	0.3	0.32	0.172	0.678
Age	1	3.8	3.81	2.042	0.153
Father’s Occupation	7	16.1	2.3	1.232	0.283
Mother’s Occupation	3	9.1	3.03	1.625	0.182
Siblings	1	0.8	0.807	0.356	0.551

**Table 4 antibiotics-14-00979-t004:** Linear mixed effect model.

Variables	Estimate	95% CI	*p*-Value
Timepoint			
Before	-		
After	1.27	1.03–1.51	<0.001
Gender			
Female	-		
Male	−0.24	−0.49–0.01	0.066
Subject			
Business	-		
Science	0.35	−0.02–0.73	0.069
Intra-Class Correlation Coefficient	0.033		

**Table 5 antibiotics-14-00979-t005:** Study Interventions.

Day 1	**Activities**	**Description of the Activities**
	Pre-survey	
	Presentation	Antibiotics can be identified as per their generics. Children do not have any knowledge about the generics of antibiotics. In 2022, DGDA has taken an initiative to incorporate “red identification marks” with the text “Antibiotic” in the packaging of antibiotics with the message “Do not take this medicine without the prescription from a registered physician”. This initiative was taken for easy identification of antibiotics and to create public awareness [11]. In the presentation, DGDA’s red label initiatives and the mechanism of antimicrobial resistance were presented.
	Comic book reading	DGDA developed the comic book “*Tinu Minu and Super Bug*” for children [32]. In this program this comic book was freely distributed among the schoolchildren, and two students were invited to read the comic book. The comic book “*Tinu Minu and Super Bug*” tells the story of antibiotic misuse through self-medication without consulting a doctor—an unfortunately common practice in many low- and middle-income countries, including Bangladesh. While children may see this as normal, the comic highlights the serious risks associated with self-medicating with antibiotics.
	Watching animation	DGDA developed an animation of the comic book “*Tinu Minu and Super Bug*” for children, which was played [33]. To capture the attention and interest of the children after reading the comic book, the animation was played.
	Coloring book reading	An antibiotic-themed coloring book developed by DGDA, “*Invention of Penicillin*”, was a black-and-white book that tells the story of Sir Alexander Fleming’s invention of penicillin [34]. Children can color the illustrations while enjoying the story, making it both educational and interactive.
	Quiz	A quiz competition was conducted, followed by a prize-giving session where students were asked simple questions about what they had learned from the interventions.
Day 2	Art competition followed by a prize-giving ceremony	On day 2, an art competition was conducted on the topic of “spread of antimicrobial resistance”. The students had already visualized the theme through the interventions.
	Post-Survey	

## Data Availability

The data pertaining to this study are shareable and available from the corresponding author upon reasonable request.

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
