# Peer review of "Assessing the Impact of Antimicrobial Resistance Awareness Interventions Among Schoolchildren in Bangladesh"

_antibiotics, 2025, doi:10.3390/antibiotics14100979_

Round 1

Reviewer 1 Report

Comments and Suggestions for Authors

This manuscript can be accepted after major revisions. Below are my comments.

1. From the socio-demographic characteristics in Table 1, the percentage disparities in gender and ethnicity appear to be quite substantial. Do these large disparities raise concerns about the fairness and generalizability of the study results? Specifically, the gender ratio shows a 2:1 male-to-female distribution, while the religious composition is heavily skewed with Islam:Hindu:Buddhist ratios of 97.5:1.7:0.8. Given these pronounced imbalances in both gender and ethnic representation, my question is whether the study findings can provide reliable guidance for broader populations.

2. The study participants are aged 12-16 years, which represents a relatively narrow age range. Do the authors have plans to expand the age range in future studies to enhance the comprehensiveness and applicability of this research?

3. While children are being taught about the dangers of antimicrobial resistance, which should theoretically lead to more judicious antibiotic use in adulthood, could this education inadvertently create the opposite extreme? Might there be a risk of patients becoming so cautious that they avoid seeking medical treatment altogether or refuse necessary antibiotic therapy even in severe cases where it is clinically indicated? This could potentially lead to a phenomenon where patients delay treatment to the point where their conditions become critical.

Author Response

Response to Reviewer 1:

Title: Assessing the Impact of Antimicrobial Resistance Awareness Interventions among School Children in Bangladesh

Manuscript ID: antibiotics-3874892
Type of manuscript: Article

Reviewer 1: This manuscript can be accepted after major revisions. Below are my comments.

Comment 1: From the socio-demographic characteristics in Table 1, the percentage disparities in gender and ethnicity appear to be quite substantial. Do these large disparities raise concerns about the fairness and generalizability of the study results? Specifically, the gender ratio shows a 2:1 male-to-female distribution, while the religious composition is heavily skewed with Islam:Hindu:Buddhist ratios of 97.5:1.7:0.8. Given these pronounced imbalances in both gender and ethnic representation, my question is whether the study findings can provide reliable guidance for broader populations.

Response 1: We appreciate the reviewer’s careful observation. We included the text below to explain Table 1: "The majority of participants were aged 14–16 years (88.4%), with males comprising nearly two-thirds (62.7%). Most belonged to class 8 (46.9%), and Islam was the predominant religion (97.5%). Fathers were commonly engaged in business (28.2%) or the armed forces (24.1%), while most mothers were housewives (75.5%). Almost half of the students (46.6%) had two siblings."

We added the text below in the discussion section: “The socio-demographic composition was imbalanced, with a higher proportion of male and predominantly Muslim participants, which may limit the generalizability of the findings.

Comment 2: The study participants are aged 12-16 years, which represents a relatively narrow age range. Do the authors have plans to expand the age range in future studies to enhance the comprehensiveness and applicability of this research?

Response 2: We added the text below in the discussion section: The age range of participants was restricted to 12–16 years, representing a critical stage for shaping health behaviors, but future studies should include broader age groups.”

Comment 3: While children are being taught about the dangers of antimicrobial resistance, which should theoretically lead to more judicious antibiotic use in adulthood, could this education inadvertently create the opposite extreme? Might there be a risk of patients becoming so cautious that they avoid seeking medical treatment altogether or refuse necessary antibiotic therapy even in severe cases where it is clinically indicated? This could potentially lead to a phenomenon where patients delay treatment to the point where their conditions become critical.

Response 3: We added the text below in the discussion section: “Although the intervention emphasized judicious antibiotic use, there is a potential risk that excessive caution could lead to underuse of antibiotics when clinically necessary, highlighting the need for balanced messaging in future programs.

Comment 4: Figures and tables can be improved

Response 4: We thank the reviewer for this valuable suggestion. We have revised the figures and tables to enhance clarity, readability, and visual presentation.

Reviewer 2 Report

Comments and Suggestions for Authors

The authors address an important and timely topic, raising awareness about antimicrobial resistance, a critical global health challenge. I appreciate their approach of engaging children through educational interventions to foster early understanding of this issue. The pilot study results are promising, and I hope such interventions will be implemented more broadly across the country to help instill the importance of combating antimicrobial resistance.

  1. I strongly believe that the written descriptions in the Results section could be substantially enriched to improve the overall quality of the paper. Currently, the section relies primarily on tables and graphs without sufficient accompanying explanation.
  2. Do the authors plan to include parents in such exercises in the future?

Author Response

Response to Reviewer 2:

Title: Assessing the Impact of Antimicrobial Resistance Awareness Interventions among School Children in Bangladesh

Manuscript ID: antibiotics-3874892
Type of manuscript: Article

Response to Reviewer 2:

The authors address an important and timely topic, raising awareness about antimicrobial resistance, a critical global health challenge. I appreciate their approach of engaging children through educational interventions to foster early understanding of this issue. The pilot study results are promising, and I hope such interventions will be implemented more broadly across the country to help instill the importance of combating antimicrobial resistance.

Comment 1: I strongly believe that the written descriptions in the Results section could be substantially enriched to improve the overall quality of the paper. Currently, the section relies primarily on tables and graphs without sufficient accompanying explanation.

Response 1: We have improved the result section, including the following explanations:

  • Table 1 shows the socio–demographic characteristics of the respondents (N=241). The majority of participants were aged 14–16 years (88.4%), with males comprising nearly two-thirds (62.7%). Most belonged to class 8 (46.9%), and Islam was the predominant religion (97.5%). Fathers were commonly engaged in business (28.2%) or the armed forces (24.1%), while most mothers were housewives (75.5%). Almost half of the students (46.6%) had two siblings.
  • Table 2 presents the paired t-test comparing pre–and post–intervention effects. Knowledge scores significantly increased after the intervention (mean difference = 1.28, p < 0.001), suggesting that the educational program was effective in improving participants’ understanding. This improvement was statistically highly significant (<0.001).
  • Figure 1 shows the percentage of correct answers by question for pre–and post–intervention, Correct responses increased across almost all questions, with the most notable improvements in items related to infection prevention and antibiotic use, indicating that the intervention addressed key knowledge gaps.
  • Table 3 shows the one–way Analysis of Variance (ANOVA), which indicates that the improvement in knowledge was consistent across gender, class, subject, religion, and parental occupation, as none of these factors showed significant differences (p > 0.05). This suggests that the intervention was effective regardless of socio-demographic background.
  • Figure 2 (a) shows that before the intervention, most participants (60.2%) had never heard of AMR. After the intervention, awareness rose dramatically, with over 93% reporting familiarity with AMR.
  • Figure 3 shows the sources from which participants learned about AMR before the interventions with school/teacher and media/advertisement being the main sources.
  • Figure 4 shows children’s illustrations revealed their grasp of how AMR spreads, with common themes including inappropriate antibiotic use, environmental contamination, and human–to–human transmission. These qualitative findings support the quantitative results by showing that the intervention enhanced both awareness and conceptual understanding.

 Comment 2: Do the authors plan to include parents in such exercises in the future?

Response 2: We added the text below in the discussion section: In addition, involving parents in future interventions could further reinforce learning at home and promote responsible antibiotic use within families, enhancing the overall effectiveness of AMR education programs.

Comment 3: The English could be improved to more clearly express the research.

Response 3: We thank the reviewer for this suggestion. We have carefully revised the manuscript to improve clarity, grammar, and overall readability. Sentences have been restructured where needed, and terminology has been standardized.

Reviewer 3 Report

Comments and Suggestions for Authors

Antimicrobial resistance is a growing concern worldwide. In developing countries, where antibiotics are accessible without a prescription, the problem of emerging antimicrobial resistance is even more prominent. In this report, the authors present the impact of education on children and adults, between 12 and 16 years, regarding the unnecessary consumption of antibiotics.  The report is well and clearly presented. Herein are some comments for the reviewers:

  1. Is the unnecessary administration of antibiotics, without a prescription, a decision of the children or of the adults? Even if the children do not want to consume antibiotics easily,  can the parents break their will and force the children to consume them?
  2. If the mother does not want to administer over-the-counter antibiotics to her children, but the father disagrees, whose is going to make the decision?
  3. Do the authors believe that both parents need to be educated regarding antimicrobial resistance?
  4. Will the authors advise the appropriate government authorities to minimize the classes and the wide-spectrum antibiotics being accessible without prescription?
  5. Do the authors believe that the antimicrobial resistance in Pakistan is increased by the inappropriate use in the hospital setting, as well? If yes, do they think that targeted training in physicians and nurses is crucial?

Author Response

Response to Reviewer 3:

Title: Assessing the Impact of Antimicrobial Resistance Awareness Interventions among School Children in Bangladesh

Manuscript ID: antibiotics-3874892
Type of manuscript: Article

Reviewer 3:

Antimicrobial resistance is a growing concern worldwide. In developing countries, where antibiotics are accessible without a prescription, the problem of emerging antimicrobial resistance is even more prominent. In this report, the authors present the impact of education on children and adults, between 12 and 16 years, regarding the unnecessary consumption of antibiotics.  The report is well and clearly presented. Herein are some comments for the reviewers:

Comment 1: Is the unnecessary administration of antibiotics, without a prescription, a decision of the children or of the adults? Even if the children do not want to consume antibiotics easily,  can the parents break their will and force the children to consume them?

Comment 2: If the mother does not want to administer over-the-counter antibiotics to her children, but the father disagrees, whose is going to make the decision?

Comment 3: Do the authors believe that both parents need to be educated regarding antimicrobial resistance?

Response 1,2 and 3:  We added the text below in the discussion section: The decision to administer antibiotics without a prescription is often determined by adults, particularly parents, rather than the children themselves. Even when children are reluctant, parental authority can lead to the use of antibiotics, highlighting the importance of involving both mothers and fathers in AMR educational programs to ensure consistent messages at home.”

Comment 4: Will the authors advise the appropriate government authorities to minimize the classes and the wide-spectrum antibiotics being accessible without prescription?

Response 4: We added the text below in the discussion section: “Educating parents alongside children can reinforce appropriate practices and reduce unnecessary or inappropriate antibiotic use. Furthermore, restricting over–the–counter access to antibiotics is supported by the Drug and Cosmetics Act 2023, which imposes a penalty of 20,000 BDT (approximately 165 USD) for selling antibiotics without a prescription in Bangladesh, helping to minimize misuse.”

Reference: The Drug & Cosmetics Act-2023–Directorate General of Drug Administration. Available online: https://dgda.gov.bd/site/view/law/%E0%A6%86%E0%A6%87%E0%A6%A8-%E0%A6%93-%E0%A6%AC%E0%A6%BF%E0%A6%A7%E0%A6%BF- (accessed on 25 September 2025)

 Comment 5: Do the authors believe that the antimicrobial resistance in Pakistan is increased by the inappropriate use in the hospital setting, as well? If yes, do they think that targeted training in physicians and nurses is crucial?

Response 5: We added the text below in the discussion section: “In addition, inappropriate antibiotic use in hospital settings also contributes to antimicrobial resistance, indicating that targeted training for physicians, nurses, and other healthcare providers is essential to address AMR comprehensively at both community and clinical levels.

Reference: Davey PG, Marwick C. Appropriate vs. inappropriate antimicrobial therapy. Clinical Microbiology and Infection. 2008 Apr;14:15-21.

Round 2

Reviewer 1 Report

Comments and Suggestions for Authors

This manuscript can be accepted.